# Transportable hyperpolarized metabolites

Xiao Ji[1], Aurélien Bornet[1], Basile Vuichoud[1], Jonas Milani[1], David Gajan[2], Aaron J. Rossini[1], Lyndon Emsley[1], Geoffrey Bodenhausen[1,3,4] & Sami Jannin[1,2]

Nuclear spin hyperpolarization of $^{13}$C-labelled metabolites by dissolution dynamic nuclear polarization can enhance the NMR signals of metabolites by several orders of magnitude, which has enabled *in vivo* metabolic imaging by MRI. However, because of the short lifetime of the hyperpolarized magnetization (typically <1 min), the polarization process must be carried out close to the point of use. Here we introduce a concept that markedly extends hyperpolarization lifetimes and enables the transportation of hyperpolarized metabolites. The hyperpolarized sample can thus be removed from the polarizer and stored or transported for use at remote MRI or NMR sites. We show that hyperpolarization in alanine and glycine survives 16 h storage and transport, maintaining overall polarization enhancements of up to three orders of magnitude.

[1] Ecole Polytechnique Fédérale de Lausanne, Institut des Sciences et Ingénierie Chimiques, Lausanne 1015, Switzerland. [2] Univ Lyon, CNRS, Université Claude Bernard Lyon 1, ENS de Lyon, Institut des Sciences Analytiques, UMR 5280, 5 rue de la Doua, 69100 Villeurbanne, France. [3] Département de Chimie, Ecole Normale Supérieure, PSL Research University, UPMC Univ Paris 06, CNRS, Laboratoire des Biomolécules (LBM), 24 Rue Lhomond, 75005 Paris, France. [4] Sorbonne Universités, UPMC Paris 06, Ecole Normale Supérieure, CNRS, Laboratoire des Biomolécules (LBM), Paris, France. Correspondence and requests for materials should be addressed to S.J. (email: sami.jannin@univ-lyon1.fr).

Dissolution dynamic nuclear polarization (d-DNP)[1,2] is a powerful method to enhance nuclear magnetic resonance (NMR) signals by several orders of magnitude, notably in [13]C-labelled metabolites. The production of [13]C-hyperpolarized metabolites has opened the way to a broad range of novel experiments, such as the detection of intermediates in fast chemical reactions[3] the observation of protein folding in real time[4] or the detection and monitoring of cancer in humans[5]. In d-DNP experiments, the [13]C metabolites are usually polarized at low temperatures ($1.2 < T < 4.2$ K) and moderate fields (usually $3.35 < B_0 < 6.7$ T) either directly[6] or indirectly[7–10] by $^1H \rightarrow {}^{13}C$ cross-polarization (CP)[11,12]. The sample formulation usually consists of a homogeneous aqueous mixture of paramagnetic polarizing agents (PAs) and metabolites sometimes containing a glass-forming agent such as glycerol. The frozen solution is then dynamically polarized by microwave irradiation. The formation of a glass upon freezing is critical for efficient DNP[13,14]. Alternatively, the PAs can be covalently attached to the surface of mesostructured materials that are impregnated with aqueous solutions of metabolites[15], in which case one can dispense with glass-forming agents. The PAs may also be generated *in situ* by ultraviolet irradiation[16]. However, intimate contact of the nuclear spins with the PA leads to paramagnetic relaxation that is exacerbated at low fields and thus requires dissolution of the sample directly in the cryostat. Hyperpolarized solutions have lifetimes $T_1(^{13}C) \sim$ 30–60 s in carboxyl groups that are sufficiently long for immediate imaging or spectroscopy, but not for transport of the sample to a distant user site.

After DNP at low temperature, the glassy mixture is rapidly melted at high field inside the DNP polarizer, and the resulting hyperpolarized solution is rapidly transferred to an NMR or magnetic resonance imaging (MRI) magnet. The hyperpolarization lifetime following dissolution is determined by the [13]C longitudinal relaxation time $T_1(^{13}C)$ in solution. The requirement of melting samples within the polarizer is one of the major limitations of d-DNP. It obviously implies that each NMR or MRI magnet needs to have a dedicated DNP polarizer located within a few metres. All attempts to remove the hyperpolarized solid sample from the polarizer, transport it and melt it remotely have so far failed. The reason for this is simply that $T_1(^{13}C)$ becomes prohibitively short in the frozen solid samples at low magnetic fields because of the close proximity of the [13]C spins to the statistically dispersed PAs[17]. To remove the polarized solid from the polarizer, the PAs and the substrate would need to be physically separated from each other, yet close enough for the PAs to polarize the substrate.

This can be achieved by preparing DNP samples with a suitable multi-component nano- or micro-particulate architecture. The radical-rich phase (RRP) contains the PAs and abundant spins such as protons, and the other radical-free phase (RFP) contains the molecules to be enhanced and the same abundant spins. The abundant protons are directly and rapidly polarized by DNP in the RRP, and this polarization is then slowly relayed to the protons of the RFP by proton–proton spin diffusion. The time needed for spin diffusion to propagate the proton polarization from the RRP to the RFP depends on the size of the RFP domains. Nano-crystalline suspensions have been polarized by DNP relayed by spin diffusion by van der Wel *et al.* in magic angle spinning studies of amyloid fibrils[18–21]. More recently, the concept of DNP relayed by spin diffusion has been generalized to micro-particulate samples by Emsley and co-workers, who showed that this could lead to substantial polarization in ordinary organic solids and provide information about domain structures, for example, in pharmaceutical formulations[22–24]. However, all of these studies were performed in combination with magic angle spinning DNP at temperatures $\sim 100$ K.

In this communication, capitalizing on this concept of DNP relayed by spin diffusion, we show how micro-particles can be hyperpolarized in trapped states with high polarization

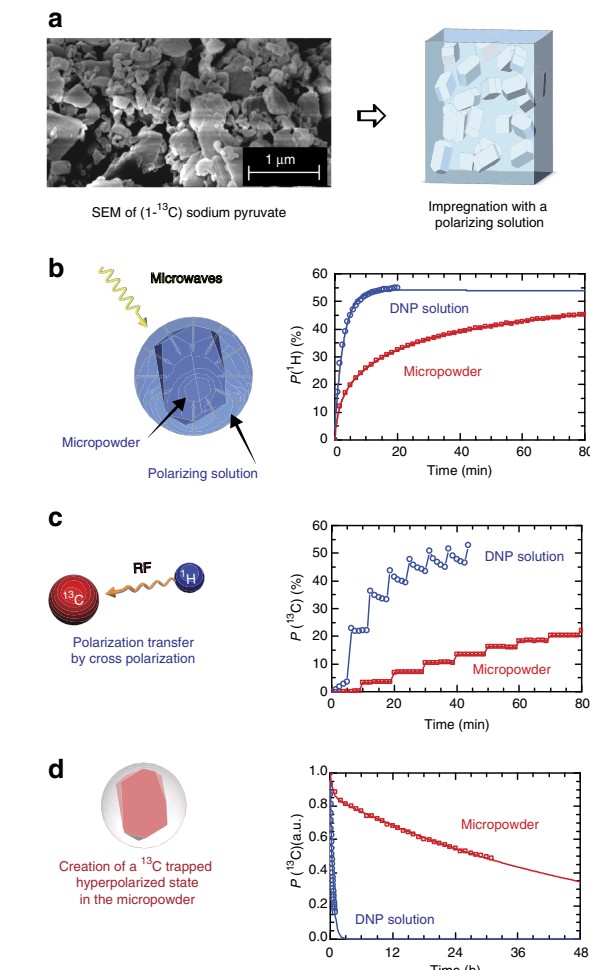

**Figure 1 | Illustration of the steps leading to a locked hyperpolarized state.** (**a**) Preparation of an impregnated powder. The radical-free phase (RFP), here a micro-powder of hydrophilic [1-[13]C] sodium pyruvate ground by hand, as shown on the SEM image, is impregnated with the radical-rich phase (RRP), here a hydrophobic glass-forming solution. (**b**) Spin diffusion relayed dynamic nuclear polarization (DNP). The resulting micro-formulation is placed in the polarizer and irradiated with microwaves. Initially, DNP enhances the proton polarization within the RRP and proton–proton spin diffusion transports the polarization to the RFP. The graph shows the [1]H DNP build-up at 6.7 T and 1.2 K in (blue circles, DNP solution) for a homogeneous frozen glassy mixture of sodium [1-[13]C]pyruvate in $H_2O$:$D_2O$:glycerol-$d_8$ (2:3:5) and doped with 40 mM TEMPOL without RFP, and (red squares, micro-powder) the build-up in micro-particles of pure sodium [1-[13]C]pyruvate (the RFP) that were impregnated with a solution of toluene-$d_6$:THF-$d_8$:THF (8:1:1) doped with 40 mM TEMPOL-benzoate as RRP. The curves show single- and bi-exponential fits to the data, as described in the text. (**c**) $^1H \rightarrow {}^{13}C$ cross-polarization (CP). Several CP pulse sequences (see methods for details on the pulse sequence) are applied to transfer the proton polarization in the RFP to carbon-13 of the metabolites. The graph shows the [13]C CP-DNP build-up curves of the same glassy RRP (blue circles) and micro-particulate RFP (red squares). (**d**) [13]C-trapped state at 4.2 K. As the carbon-13 nuclei in the RFP are not in close contact with the paramagnetic radicals of the RRP, the hyperpolarized [13]C magnetization has a long lifetime $T_1(^{13}C)$ at 4.2 K. The graph shows the decay of the [13]C nuclear spin polarization at 4.2 K and 6.7 T in the same glassy RRP (blue circles) and micro-particulate RFP (red squares).

levels of typically $P(^{13}C) > 10\%$, and with long lifetimes at 4.2 K of typically $T_1(^{13}C) > 5$ h. We also show that due to the special nature of these states, the polarized sample can be removed from the DNP polarizer while preserving much of its polarization, in contrast to $^{13}C$ spins of metabolites dissolved in conventional glassy DNP samples and similar to silicon nanoparticles[25,26]. The polarized sample can even be transported, and finally dissolved to the liquid state in a remote location. If the RRP is formulated to be hydrophobic, the hyperpolarized metabolites (that are invariably hydrophilic) can be easily separated from the PAs and any other contaminants during the dissolution process, yielding a pure hyperpolarized solution without the need for further purification[6,27–31].

## Results

**Overview.** We show here that protons in particles of metabolites with diameters of a few micrometres are polarized at $T = 1.2$ K and $B_0 = 6.7$ T on a timescale of 30 min by DNP-assisted proton–proton spin diffusion. CP is then used to transfer the polarization from the $^1H$ in the RFP to the $^{13}C$ spins of the metabolites. After the $^{13}C$ polarization has achieved a satisfactory level, the microwave irradiation is switched off and the sample temperature is increased from 1.2 to 4.2 K (ambient pressure), and the protons in both RRP and RFP phases relax towards thermal equilibrium within minutes. However, the $^{13}C$ spins are effectively isolated from both protons in the RFP and PAs in the RRP, and therefore can remain hyperpolarized for prolonged periods, typically for days at 4.2 K in a field of 6.7 T. Indeed, at this point, the $^{13}C$ hyperpolarization is trapped in the RFP because, first, there are no PAs acting as relaxing agents and, second, the $^{13}C$ hyperpolarization cannot efficiently diffuse out of the RFP, since $^{13}C–^{13}C$ spin diffusion is intrinsically two orders of magnitude slower than $^1H–^1H$ spin diffusion in the RFP, and is ineffective in the RRP since it is not isotopically $^{13}C$ enriched.

**Sample formulation.** RFP of [1-$^{13}C$]Glucose, [$^{13}C_3$, $^{15}N$]alanine, [$^{13}C_2$, $^{15}N$]glycine, and sodium [1-$^{13}C$]pyruvate were chosen as test substances. Sodium [1-$^{13}C$]pyruvate has become the metabolite of reference for *in vivo* clinical studies, where it provides a promising means to detect prostate tumours[5]. More generally, any substrate that can be prepared in the form of a micro-particulate powder is equally suitable for our method. Here

powdered samples were ground by hand to an average particle diameter of $1 < d < 10 \mu m$, as determined by scanning electron microscopy (SEM; Fig. 1a). Smaller particle diameters reduce the distance over which hyperpolarized $^1H$ magnetization needs to diffuse and accelerate the build-up rates.

The RRP consists of a glass-forming solution of PA chosen such that the RFP is not soluble in the RRP, such as 80 mM TEMPOL-benzoate in toluene/THF (v:v = 8:2; see Methods for details on wet impregnation of the RFP with the RRP, including procedures to homogenize the distribution, and to fix the sample by freezing). We have not yet investigated bio-compatible solvents that could replace toluene. This DNP medium was chosen because, first, a proton polarization $P(^1H) > 50\%$ can be achieved in $< 5$ min at 1.2 K and 6.7 T (Fig. 1b), second, none of the metabolites that we have tested are soluble in this mixture and, third, since the mixture is not soluble in water, it separates after dissolution. Other combinations of frozen glass-forming solvents and PA's are possible, and may turn out to be more effective.

**Spin diffusion relayed DNP.** The blue curve in Fig. 1b shows the $^1H$ DNP build-up at 1.2 K and 6.7 T for a typical RRP, a homogeneous frozen glassy solution made by dissolving sodium [1-$^{13}C$]pyruvate in $H_2O:D_2O:$glycerol-$d_8$ (2:3:5) doped with 40 mM TEMPOL, without any micro-particles containing metabolites (that is, without any RFP). The polarization reaches $P(^1H) > 50\%$ with a mono-exponential build-up with a characteristic time constant $\tau_{DNP}(^1H) = 150$ s. The red curve in Fig. 1b shows the $^1H$ DNP build-up obtained under the same conditions by wet impregnation of 20 mg of micro-crystalline sodium [1-$^{13}C$]pyruvate RFP powder (proton density of 34 m mol cm$^{-3}$) with 60 μl of toluene-$d_6$:THF-$d_8$:THF (8:1:1) doped with 40 mM TEMPOL-benzoate as RRP (proton density of 9.4 m mol cm$^{-3}$). The $^1H$ DNP build-up curve is now bi-exponential with a fast component $\tau_{DNP}{}^{fast}(^1H) = 155$ s arising from the direct polarization within the RRP, and a new slow component $\tau_{DNP}{}^{slow}(^1H) = 1900$ s corresponding to polarization of the RFP that builds up through proton–proton spin diffusion across the phase barrier between the RRP and the RFP. This clearly shows that the polarization enhanced by DNP propagates, albeit slowly, from the RRP to the RFP.

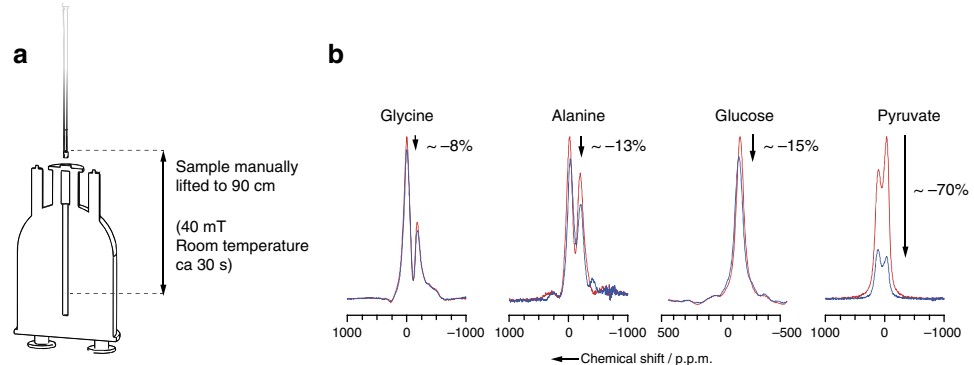

**Figure 2 | Preserving hyperpolarization out of the polarizer.** (**a**) Procedure for lifting the sample above the polarizer and putting it back. In practice, the sample is lifted manually by 2 cm into a coil that provides a supplementary field of 40 mT, parallel to the $B_0$ field of the polarizer. In 5 s, the sample and the supplementary coil are then lifted together 90 cm above the centre of the main $B_0$ field. The sample is kept at room temperature for ~30 s and is put back into the polarizer in ~5 s. (**b**) $^{13}C$ NMR spectra before and after lifting the solid powder out of the polarizer. $^1H$ decoupled $^{13}C$ NMR of [$^{13}C_2$, $^{15}N$]glycine, [$^{13}C_3$, $^{15}N$]alanine, [1-$^{13}C$]glucose and [1-$^{13}C$]pyruvate measured with 0.5° nutation angle pulses after 10 min $^1H$ DNP followed by a single $^1H$–$^{13}C$ cross-polarization contact, before (red line) and after (blue line) lifting the sample according to the procedure described in **a**, leading to polarization losses of ca. –8%, –13%, –15% and –70%, respectively, illustrated by the arrows.

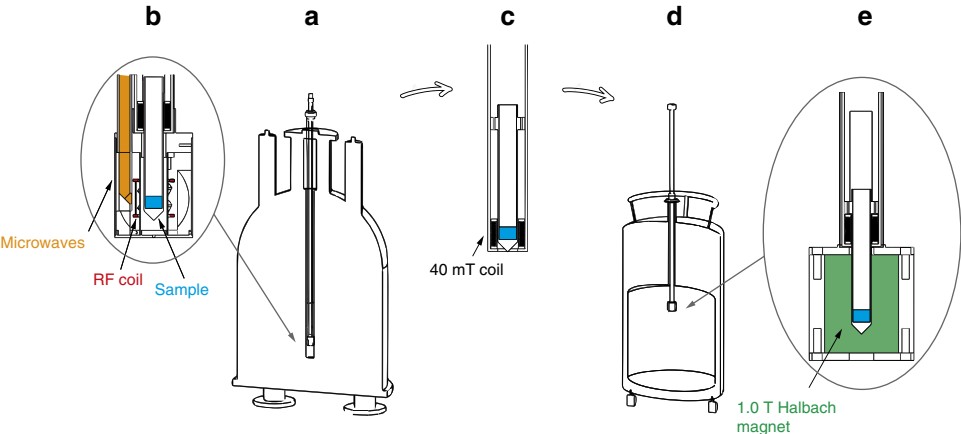

**Figure 3 | Illustration of the remote hyperpolarization set-up. (a)** The DNP polarizer consists of a 6.7-T wide bore magnet and a 1.2-K cryostat equipped with (**b**) a DNP probe where the sample is inserted and irradiated with microwaves for $^1$H DNP and radio-frequency fields for $^1$H–$^{13}$C cross-polarization. The sample can then be manually removed from the polarizer in (**c**) a transfer stick comprising a coil sustaining a magnetic field of ca. 40 mT (100 turns, current of 4 A) and subsequently inserted in (**d**) a conventional liquid helium transport Dewar with (**e**) a magnetic insert providing a 1.0 T static magnetic field for storage or transport.

The DNP enhanced proton polarization can then be transferred to the $^{13}$C spins in the RFP by $^1$H → $^{13}$C CP at low temperatures[7,8,11,12,32]. Although our micro-particulate DNP sample formulation is drastically different from the usual glassy solutions, $^1$H to $^{13}$C by CP-DNP was performed in essentially the same manner. Figure 1c shows the stepwise build-up of the $^{13}$C magnetization of sodium [1-$^{13}$C]pyruvate obtained by multiple-contact CP from $^1$H to $^{13}$C at 1.2 K, compared with a similar $^1$H → $^{13}$C CP build-up measured on the same amount of sodium [1-$^{13}$C]pyruvate, but dissolved in the conventional glassy matrix H$_2$O:D$_2$O:glycerol-d$_8$ (2:3:5) doped with 40 mM TEMPOL[33]. Though these two $^1$H → $^{13}$C CP build-up curves look similar, it is worth highlighting two major differences, first, the time between two consecutive CP contacts needs to be extended to allow proton–proton spin diffusion to carry the polarization into the RFP particles between CP steps ($\Delta t^{CP} = 20$ min in the two-component sample, instead of 5 min in the conventional glassy matrix) and, second, $^{13}$C relaxation between CP contacts, which usually limits the $^{13}$C build-up, is essentially absent in the micro-particulate RFP because of the absence of contact with the PAs. These two new features imply that the $^{13}$C polarization is slower to build-up, but can in principle achieve polarization levels as high as $P(^{13}C) = P(^1H)$.

**Remote hyperpolarization.** *Extended hyperpolarization lifetimes.* The primary novelty associated with this new sample formulation resides in the markedly different $^{13}$C spin-lattice relaxation times $T_1(^{13}C)$. The $^{13}$C spins of metabolites dissolved in glassy matrices are inevitably in contact through electron–nuclear dipolar interactions with the PAs that act as paramagnetic relaxation centres. In the micro-particle formulation on the other hand, the $^{13}$C spins of the RFP are physically separated from the PAs on a micrometre length scale that is much larger than that of the electron–nuclear dipolar interaction. Figure 1d shows how the spin-lattice relaxation time $T_1(^{13}C)$ of sodium [1-$^{13}$C]pyruvate can be markedly extended at 4.2 K and 6.7 T by switching from a glassy frozen solution, where $T_1(^{13}C) = 20$ min, to the micro-particulate sample, where $T_1(^{13}C)$ is extended to 37 h. We measured relaxation times exceeding 20 h in [1-$^{13}$C]glucose, and 5 h in [$^{13}$C$_3$, $^{15}$N]alanine and [$^{13}$C$_2$, $^{15}$N]glycine.

*Sample transfer and storage.* The paramagnetic-free environment in the RFP enables a marked extension of the nuclear spin-lattice relaxation time $T_1(^{13}C)$. Transport to remote locations becomes possible provided that one uses a simple cryogenic transport device maintaining a small static magnetic field. This was demonstrated recently in the context of brute force hyperpolarization where neat [1-$^{13}$C]pyruvic acid was thermally polarized without PAs by cooling to 2 K in a field of 14 T (refs 34,35). Special care needs to be taken with regard to the magnetic field for sample shuttling out of the polarizer, so as to avoid excessive relaxation losses and low-field nuclear thermal mixing[36,37] as recently illustrated by Peat *et al.*[26] To test the feasibility of our strategy towards transporting hyperpolarization, Fig. 2 shows how lifting the sample manually above the polarizer for 30 s in a 40 mT magnetic field nicely preserves the hyperpolarization for [1-$^{13}$C]glucose, [$^{13}$C$_3$, $^{15}$N]alanine or [$^{13}$C$_2$, $^{15}$N]glycine and to a lesser extent for sodium [1-$^{13}$C]pyruvate. For pyruvate, the manual transfer strategy is not satisfactory, and a faster automated, cold, higher field transfer would help[34,35] overcoming the fast relaxation arising from the presence of methyl groups and quadrupolar sodium nuclear spins in the vicinity of $^{13}$C.

The feasibility of transfer and storage is demonstrated for a mixture of [$^{13}$C$_3$, $^{15}$N]alanine and [$^{13}$C$_2$, $^{15}$N]glycine powders impregnated with the RRP. For that purpose, we have upgraded our existing state-of-the-art DNP polarizer with a transfer, storage and transport system, as shown in Fig. 3. The sample was polarized at 1.2 K in our polarizer (Fig. 1a,b) for 60 min, while three CP sequences were applied at 20, 40 and 60 minutes, leading to a polarization level $P(^{13}C) \sim 15 \pm 5\%$. The sample was manually transferred out of the 6.7 T polarizer in a 40 mT magnetic field (Fig. 3c) and inserted into the magnetic insert of the cryogenic Dewar (1.0 T and 4.2 K; Fig. 3d,e) and stored for 16 h. No attempts were made so far to optimize the $^{13}$C lifetimes during the waiting time of 16 h. Higher fields, lower temperatures, sample purification, oxygen removal and annealing of the RFP are expected to improve the lifetimes, as shown by Kempf *et al.*, who reported $T_1(^{13}C) > 24$ h at 6 K and 4 T in neat pyruvic acid after annealing[34,35]. In this case, we used the sample without purification for this proof-of-concept experiment.

*Dissolution.* Following a 16-h hold, the sample was then dissolved directly from the transport Dewar following a previously described procedure[9] with 5 ml of pressurized heated water (420 K and 1 MPa). The resulting hyperpolarized solution

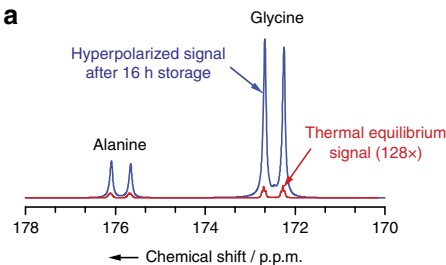

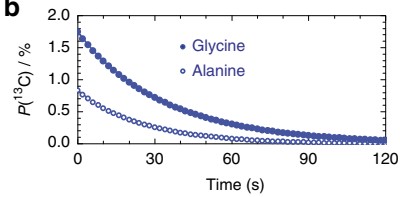

**Figure 4 | Hyperpolarization preserved during a 16-h storage.**
(**a**) Hyperpolarized 1-$^{13}$C signal in [$^{13}$C$_3$, $^{15}$N]alanine or [$^{13}$C$_2$, $^{15}$N]glycine. Before storage at 4.2 K and 1.0 T, 5.5 mg [$^{13}$C$_3$, $^{15}$N]alanine and 9 mg [$^{13}$C$_2$, $^{15}$N]glycine were impregnated with 45 µl of toluene:THF (8:2) doped with 80 mM TEMPOL-benzoate (RRP) and polarized at 6.7 T and 1.2 K. Following a 16-h hold, the sample was dissolved in the storage Dewar in 700 ms with superheated water (5 ml at 420 K) and pushed through a magnetic tunnel[38] towards the 500-MHz spectrometer in 4.5 s, and injected in a 5-mm NMR tube in 2 s. This is to be compared with a thermal equilibrium signal (red line, 128 ×) measured after complete relaxation of the $^{13}$C magnetization, with 16 pulses with 90° angles applied every 300 s. (**b**) Hyperpolarization decay after dissolution to room temperature. The complete 1-$^{13}$C signal relaxation curve was measured with 5° nutation angle pulses applied every 2 s for [$^{13}$C$_3$, $^{15}$N]alanine (blue circle) and [$^{13}$C$_2$, $^{15}$N]glycine (blue filled circle).

was pushed through a magnetic tunnel[38] and injected into a 5-mm sample tube in a 500-MHz liquid-state NMR spectrometer. Figure 4a shows the hyperpolarized signal compared with thermal equilibrium, featuring enhancement factors $\varepsilon_{DNP} = 821$ and 1728, corresponding to polarizations $P(^{13}C) = 0.84\%$ and 1.76% for alanine and glycine, respectively. In the dissolution product, we measured $^{13}$C spin-lattice relaxation times $T_1(^{13}C) = 26.3$ and 34.9 s for alanine and glycine, respectively, which are typical for radical-free solutions (Fig. 4b). This is explained by the fact that the PA used (TEMPOL-benzoate) is insoluble in water and therefore remains in the hydrophobic organic RRP phase that separates[28] from the aqueous phase, while the alanine and glycine readily dissolve in the aqueous phase. With our micro-particulate sample formulation, the thermodynamics of dissolution is advantageous compared with water-based glassy solutions since, first, the heat capacity $C_P$ and latent heat of fusion ($L_V$) of organic solvents are significantly smaller than those of water and, second, the enthalpy of solution ($\Delta H_{soln}$) of organic crystals is generally negative, so that dissolution is exothermic. Dissolution can therefore be performed with the existing dissolution devices[6,9,39,40], and possibly with a smaller dilution factor.

## Discussion

This proof-of-concept experiment is not yet optimized, and the levels of polarization obtained after transfer, storage and dissolution are about one order of magnitude below what can

be achieved in conventional 'on-site' d-DNP in frozen glassy samples. The reasons for the lower polarization are multiple.

First, DNP would be more efficient, typically by a factor two, if the particles were tailored to the right dimensions. Indeed, small particle diameters are critical to reduce the distance over which hyperpolarized $^1$H magnetization needs to diffuse. Zirconia ball milling or other preparation methods could be envisaged, such as precipitation/crystallization from a saturated solution, co-crystallization, spray-drying, re-crystallization by solvent evaporation and so on.

Second, polarization losses above 10% are currently observed upon transfer of the sample to the cryogenic storage Dewar. This could be minimized if the transfer were done more rapidly, at a lower temperature and in a higher magnetic field.

Third, losses during storage and transport could possibly be minimized if the storage magnetic field were increased. As shown in Fig. 1d, polarization losses in sodium [1-$^{13}$C]pyruvate at 4.2 K and 6.7 T are only ~40% over 16 h.

Finally, sample formulation could be further improved by a series of measures among which removal of traces of paramagnetic ions with suitable ligands, purification by re-crystallization, annealing, and deoxygenation to promote longer hyperpolarization lifetimes.

In conclusion, we have introduced a novel approach to dissolution DNP that uses samples with a micro-particulate architecture where substrates such as metabolites are physically separated from the PAs, the former in an RFP, the latter in an RRP. This enables the creation of 'trapped' hyperpolarized states by a combination of proton DNP in the RRP, proton–proton spin diffusion across the phase separation between the RRP and the RFP, and CP from $^1$H to $^{13}$C within the RFP. The lifetime $T_1(^{13}C)$ of the hyperpolarization $P(^{13}C)$ in the micro-particulate samples studied here ranges between 5 and 37 h. This approach enables the removal of a solid hyperpolarized by DNP from the polarizer, while preserving its polarization. This is reminiscent of samples that have been hyperpolarized by so-called brute force methods (that is, at much lower temperatures without radicals[32,33]), although DNP can be much faster and can achieve much higher polarization levels. We have demonstrated that one can store and transport hyperpolarized molecules to remote locations using a simple cryogenic transport device. Finally, we have shown how the micro-particulate samples can be dissolved in hot water so that the aqueous RFP is physically separated from the organic RRP. The resulting pure hyperpolarized solution of metabolites can in principle be used in any existing hyperpolarized *in vivo* or *in vitro* magnetic resonance experiment. With this new approach, the production of hyperpolarized molecules could in principle be scaled up, using dedicated high-throughput multiple-sample[25] remote polarizers.

## Methods

**Sample preparation.** [1-$^{13}$C]Glucose (Sigma Aldrich), [$^{13}$C$_3$, $^{15}$N]alanine, [$^{13}$C$_2$, $^{15}$N]glycine and sodium [1-$^{13}$C]pyruvate (Cambridge Isotopes) were ground by hand to obtain the RFP. The RRPs were prepared by dissolving 4-hydroxy-TEMPO-benzoate (Sigma Aldrich) in mixtures of toluene-d$_6$ (Armar Chemicals), THF-d$_8$ (Armar Chemicals), toluene and THF (Sigma Aldrich). The finely ground RFP were then impregnated with the RRP solution directly in the DNP sample holder with 3 µl of the RRP added per milligram of RFP. The impregnated mixture in the DNP sample holder was then sonicated for 10 s before insertion in the DNP polarizer that resulted in rapid freezing of the sample in liquid helium.

**Remote DNP apparatus.** The system builds on our previously described DNP apparatus[9] that consists of a 6.7-T DNP polarizer equipped with a CP-DNP probe (Fig. 3a,b) in which the sample changer tube (Fig. 3c) has been modified to accommodate a solenoidal coil (100 turns) so as to sustain a ca. 40 mT magnetic field during sample transfer (with 4 A current). After DNP was completed, the sample was manually transferred in ca. 10 s from the 6.7-T polarizer to a 1.0-T Halbach magnet in a liquid helium cryogenic transport Dewar (Fig. 3d,e).

The polarized [$^{13}C_3$, $^{15}N$]alanine and [$^{13}C_2$, $^{15}N$]glycine powders were left in the storage Dewar for 16 h. Dissolution then took place directly in the storage Dewar.

**DNP experiments.** DNP was performed at $B_0 = 6.7$ T and $T = 1.2$ K by frequency modulated microwave irradiation ($P_{\mu W} = 87.5$ mW and $f_{\mu W} = 188.3$ GHz) and combined with CP from $^1H$ to $^{13}C$ in the RFP, using CP contacts of 1–10 ms and $B_1 = 20$ kHz at intervals of 20 min to allow the proton polarization to migrate by spin diffusion from the RRP to the RFP. Further details are available in Fig. 5a of ref. 41.

**Dissolution experiments.** Dissolution was performed as described elsewhere[39] by injecting 5 ml of $D_2O$ preheated to $T = 420$ K at 1.0 MPa onto the powdered impregnated materials in 700 ms. The dissolved sample was then pushed in 4.5 s by helium gas with a pressure of 0.6 MPa through a 1.5-mm inner diameter polytetrafluoroethylene tube protected by a 0.9-T magnetic tunnel[38] to a home-built injector just above a 5-mm sample tube in a 11.7 T magnet (500 MHz for protons). The sample was then injected in ca. 2 s. The complete sequence of dissolution, transfer and injection takes $\sim 7.2$ s. A series of 5° pulses was then applied to the $^{13}C$ nuclei at regular intervals (2–5 s).

**Scanning electron microscopy.** SEM was performed with a FEI XLF-30 SEM instrument at 10 kV accelerating voltage. An amount of 10 mg of finely ground RFP was placed on a film of conducting polymer attached to the sample holder. The sample was coated with 25 nm gold to ensure surface conductivity.

**Data availability.** The data that support the findings of this study are available from the corresponding author upon reasonable request.

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

## Acknowledgements

We thank Dr Pascal Miéville for valuable assistance. S.J. thanks James Kempf for invaluable discussions and Roger Mottier from the mechanical workshop. SEM pictures were recorded by Grégoire Baroz and Julien Michellod. This work was supported by the Swiss National Science Foundation (SNF), the Ecole Polytechnique Fédérale de Lausanne, Bruker BioSpin Switzerland AG, the Centre National de la Recherche Scientifique, Equipements d'Excellence (EQUIPEX) Contract ANR-10-EQPX-47-01, and the European Research Council (ERC) Advanced Grants 320860 and 339754.

## Author contributions

J.M. and S.J. built the apparatus. A.B., X.J., B.V., D.G., A.J.R. and S.J carried out the experiments. L.E., G.B., A.B. and S.J. conceived the ideas, analysed the data and wrote the paper.

## Additional information

**Competing financial interests:** The authors declare no competing financial interests.

