## [Peer Review File · Nature Communications]

Reviewers' comments:

Reviewer #1 (Remarks to the Author):

This manuscript presents and demonstrates an innovative, potentially transformative method for the key emerging field of metabolic imaging of hyperpolarized nuclear spins. In 'conventional' applications in this field, hyperpolarization is typically generated by dynamic nuclear polarization (DNP) using a cryogenic, microwave polarizer that is situated very close to the imaging scanner. Dissolution of the sample from the cryostat, and then injection to a patient, allow immediate imaging. A desirable, but not previously possible variation, would be to instead remove several hyperpolarized samples from the polarizer in the frozen solid state, and then store them for later use and, especially, deliver them to far away imaging centers. That would avoid the complexity and cost of co-siting cryogenic polarizers and imaging scanners, thus bringing hyperpolarized metabolic MRI to a much broader base of researchers, practitioners and patients.

Before now, it has been thought that removing frozen solid samples from a DNP polarizer would not be possible, because DNP requires the presence of free-radical electrons. The radicals cause rapid loss of hyperpolarization in the otherwise slowly relaxing solid state. Here, the authors' innovated by preparing samples with microscale separation of two components: (a) the radical-embedded DNP medium, and (b) micro-crystallites of the metabolic compound to be hyperpolarized. Not only do the authors show that large hyperpolarization (~10%, or near 10,000x enhancement) can be achieved in several compounds of interest, they also demonstrate low-loss removal from the polarizing cryostat, and then more than a day or more of storage before eventual dissolution and magnetic resonance observation. This proof of concept is sure to inspire new approaches in this exciting field, and may also enable new applications (e.g., one possibility, multiple-injection hyperpolarized imaging from several hyperpolarized samples)

I am enthusiastic about publication of this work in Nature Communications. It meets the bar for high impact and broad interest. It will be valuable not only to the very large field of metabolic and cancer imaging, but also for applied spectroscopists, who might utilize the approach in studies of (e.g.) drug binding, protein-protein interactions, etc..., where one might simply order hyperpolarized substrates to enable a new kind of experiment on already widely used NMR equipment.

While I believe this paper should ultimately be accepted to Nature Communications, I feel the authors must first address several concerns and clarifications. Significant issues follow here, after which I note minor issues.

Significant Questions / Changes:

(1) The authors should better comment on why the glycine/alanine mixture was chosen for the ultimate storage/transport experiment (Figure 4) as opposed to the other cases tested for removal from the polarizer (Figure 2). This is especially important since the authors had earlier (page 5, lines 17-121) emphasized the particular importance of pyruvate for imaging applications and gave it full focus in Fig 1 to explain and motivate the whole method. Also, they argued the CCM/CPA method should be general to any microcrystallite-forming molecule, →... based on Figure 2, it's clear that losses on removal were largest for pyruvate and smallest for Ala/Gly. So the motivation is pretty obvious. And yet explanation is still needed, including comments on why pyruvate suffered so much on removal from the cryostat.

(2) Starting on page 5, line 124, the authors refer to 'impregnation of the CCM by the CPA. This is misleading usage that should be dropped. The CCM is not impregnated by the CPA. Rather, isolated islands of CCM float in the CPA, and so it would be more accurate to say the CPA is impregnated by isolated CCM domains.

... the noted usage is consistent in the manuscript (e.g., page 6, lines 137-9, Figure 1 caption, and many others). So please search and correct all instances together.

... I suspect the choice of this language is historical, in that the wetting method for CPA / CCM combination has been referred to as 'impregnation' when applied in other work with porous radical materials. There, impregnation made sense, but the present approach is physically distinct.

(3) In Conclusions, the authors state "This approach enables for the first time the removal of the hyperpolarized solid from the polarizer while preserving polarization, and we have demonstrated for the first time that storage or transport of hyperpolarized small molecules to remote locations using a simple cryogenic transport device is possible."

... This is not entirely true and the authors need to be more careful. Both noted firsts are only true if qualified for hyperpolarization "produced by DNP". Elsewhere, the authors cited Hirsch, et al, JMR 2015, who hyperpolarized and transported pyruvic acid using the brute-force method. The authors must be equally clear in the Conclusions section. There is no reason to avoid the comparison there, i.e., it is clear that their enhancements are so far much bigger with DNP than using brute force. And so the actual 'firsts' of this paper are still high impact.

... This Conclusion must be rewritten more openly, and also with argument as to why this DNP approach may be best.

(4) The title "Remote Hyperpolarization" lacks some meaning. I would prefer something more descriptive, such as "Transportable Hyperpolarization from Remote Dynamic Nuclear Polarization".

(5) Comment: From Fig 1b, it appears that only about 15 min is needed for build-up to about 50% of the maximum achievable by traditional DNP of a homogeneous frozen solution. And then, another ~1 - 1.5 hrs are needed to get the final 50%. It would be interesting to know if the hyperpolarization obtained in the initial ~15 min has a similar or much shorter lifetime than the ultimate hyperpolarization obtained after 1-1.5 hrs.

(6) In discussion the CP + spin diffusion method to buildup ^{13}C hyperpolarization (page 6, lines 160-1), the authors wrote that two features of the method [(i) the longer time needed between CP contacts, and (ii) the (very) low loss during that time] imply that ^{13}C polarization is slower to build-up, but can in principle achieve higher final polarization levels, theoretically up to $P(^{13}\text{C}) = P(^1\text{H})$.

... However, I don't see how the data supports that the final claim is distinct over prior methods. In Fig 1c, the traditional method (without CCM isolation) yielded a sawtooth buildup of $P(^{13}\text{C})$ that appears to asymptotically approach $P(^{13}\text{C}) \sim 50\text{-}60\%$. That is very similar to the ultimate ^1H proton polarization that was shown in Fig. 1b. So it seems that the limit $P(^{13}\text{C}) = P(^1\text{H})$ can be obtained either with or without the new innovation.

... The real key (properly noted by the authors and clear from their very good Fig 1c) is that the losses appear to be negligible in the CCM/CPA case, whereas they are rapid and large for DNP of a traditional formulation. It seems an overstatement to say that higher $P(^{13}\text{C})$ could be obtained using the CCM/CPA mixture. Better to say that the new method is "equally capable of ultimately achieving the $P(^{13}\text{C}) = P(^1\text{H})$ limit."

(7) A comment: when reading results, I immediately asked, "Can a faster buildup be obtained by using smaller CCM crystallites?" Much later (lines 276-280 in discussion), the authors take on this point and suggest several approaches to adjust particle size and possibly improve throughput. This is key discussion, and an earlier mention might be helpful. Please consider as part of your revisions.

(8) The authors should also discuss another aspect of production throughput. That is, DNP is still a mostly one-sample-at-a-time method. Will that hold back development of a distributed system to deliver multiple 'doses' of hyperpolarization? What is the ultimate potential for multi-sample high throughput?

... Citations to Matthias Ernst and to GE's SpinLab as precedents for multisample production should be given. Elsewhere, they cite GE, but not Krajewski ... Ernst, Kozerke, "A multisample dissolution DNP system for serial injections in small animals.", *Mag.Res.Med.* (22 Feb 2016).

Specific minor concerns / corrections:

- page 3, lines 64-65. Where authors write, "However, in all these cases paramagnetic relaxation in low fields precludes remote d-DNP as described below", which would be improved by two main changes:

(1) Please be a more explicit here, rather than waiting to the end of next paragraph for explanation. E.g., instead use, "However, in all these cases, intimate contact of the nuclear spins with the PA yields paramagnetic relaxation that... "

(2) This sentence should not end the paragraph, and would be better as the lead of the next paragraph. Along with that move, there should be some editing of the text for content and proper flow. Specifically,

(a) dropping "precludes remote DNP" and instead writing that paramagnetic relaxation "requires dissolution of the sample directly in the cryostat."

'Remote DNP' is not really precluded, but rather is the authors' solution to the fact that the relaxation would normally preclude removal of the sample in the solid state.

(b) adding, "The resulting hyperpolarized solution has adequate lifetime ($T_1(13C) \sim 30 - 60s$ in carboxyls) for immediate imaging or spectroscopy, but not for transport of the sample to a distant user site."

... and then there will be minor adjustments to remainder of the paragraph to avoid redundancies.

- page 4, line 77. Drop the word "would". The unnecessary subjunctive case weakens the authors' statement.

- page 4, lines 79-80. There is an awkward break from traditional methods, to the method of the authors. This is easily fixed by adding the word 'here' and using active voice, as in "However, here we achieve this by preparing DNP samples with a suitable multi-component..."

- page 4, lines 87-88. Change to "Here, we show that particles with diameters on the order of a few μm typically result in a timescale of 30 minutes for proton-proton spin diffusion."

... The original combined "in a timescale with "on the order of", which is an unnecessary double qualification.

... Also, "timescale of 30 min" seems fair (e.g., time constant of 1900 s noted on line 141) and better quantifies for the reader.

... Finally, the authors might comment if they expect any field or temperature dependence for this timescale given that polarizer conditions may vary.

- page 4, line 89. Change to "...from $1H$ spins in the CCM to $13C$ spins of the metabolites."

- page 4, line 90. The authors wrote, "At this point, DNP is switched off, and...", It is not clear if they mean that DNP are switched off before or just after cross polarization. This should be clarified. Also, in Supporting Information, experiment timelines should be included, e.g., one pulse sequence (rf + microwaves + diffusion delays, etc...) should be given corresponding to Figures 1b and 1c.,

- page 4, line 92. Typical should not be used without defining the corresponding conditions. Please change "...can remain polarized, typically for days." to "..., typically for days when at polarizer conditions of 6.7 T, and 1.2 K."

- page 4, line 96. Change "... $13C$ - $13C$ spin diffusion is ineffective in the CPA." to "... $13C$ - $13C$ spin diffusion is ineffective in the natural-isotopic-abundance CPA." This is not essential, but it does make it easier for the reader to immediately grasp the reason.

- page 4, line 97. Consider using "spin-diffusion assisted, relayed DNP"

- page 5, line 103. "magic angle spinning DNP (MAS)" to "magic angle spinning (MAS) DNP"
- page 5, line 121-22. Add reference to SEM and/or other method used, e.g., "Here, powdered samples were ground by hand to an average particle diameter of $1 < d < 10 \mu\text{m}$, as determined by SEM. (See Figure 1a.)"
- On page 5, line 123, the CPA description should start by just giving the composition of the CPA. As written, It was strange to start a section on CPA with "The CCM made of micro-particles is impregnated by a CPA consisting of...". Also, see earlier note on 'impregnation'.
- On page 6, line 126, The authors should expand "(see methods)" to "(See methods for details on combining CCM and CPA, including procedures to homogenize the distribution, and to fix the sample by freezing.)". The later bit on fixing is not currently present, but would be a simple, useful addition.
- page 6, lines 148-9. Change to "was performed here" instead of "can be performed" in "..., cross-polarization from 1H to 13C by CP-DNP was performed in essentially the same manner."
- page 6, lines 137-9. Where it states, "The red curve in Figure 1b shows the 1H DNP build-up obtained under the same conditions by impregnating 20 mg of the microcrystalline sodium [1-13C]pyruvate CCM sample with the CPA."
 ... The authors should specify the fractional volume contributions of CPA and CPM, as well as the proton densities in each. This can later be pulled out of Methods, but would be better also given here.
 ... and again, fix 'impregnation'.
- page 8, Figure 1.
 ... The labeling text in 1b should be updated to match acronym definitions, i.e., change to "component containing polarizing agents (CPA) and "component containing metabolites (CCM)"
 ... In caption for (b), state "Curves are single and bi-exponential fits to the data, as described in the text."
 ... In caption for (c), add reference at "several CP pulse sequences are applied[REF] to transfer..."
 Also, please refer to sequences newly provided in Supp. Info, as requested above.
- page 9, line 184. Add "In contrast," to "In contrast, the 13C spins of metabolites dissolved in glassy..."
- page 8, Figure 2, and corresponding discussion.
 ... On page 10, line 211-14, the authors wrote that "Special care needs to be taken with regard to the magnetic field trajectory for sample shuttling out of the polarizer... Figure 2 shows how a manual sample lift above the polarizer..." The whole intent of Fig 2 results would be clearer if the second sentence started with "In order to test the feasibility of this first step towards transportable hyperpolarization, Figure 2 shows..."
 ...The 2nd sentence of the caption to part (a) is confusing. It should be broken into at least two sentences, with some clarifying changes. E.g., change to "In practice, over a period of about 15s, the sample is lifted up (____) 40 mT supplementary field coil (____) 90 cm above the center of the main B0 field. The supplementary field is in the same direction as B0 field and limits sample exposure to low field."
 The first and second blanks (____) should be filled with either "while within a" and "to a distance of", OR "and into" and "located at". That depends on whether the 40 mT coil moved with the sample OR if the sample moved into it. I assume the former, but if not, what was the minimum exposure while moving?
 It seems the 40 mT coil here was probably the same one noted in Fig 3(c), but that is not clear with Fig. 2. If the same, then the authors should modify Fig 2 to incorporate Fig. 3(c) as an inset

- and also describe it in the caption. (They should also still keep this picture in Fig.3(c).)
- ... The authors also wrote "..., where the sample is in a room temperature environment, and is put back in the polarizer in approximately the same amount of time."
- please add "for about 30 s" to the first phrase and change the last phrase to "in another ~15 s."
 - Is there a special reason to come to room T, rather than colder, polarization-preserving environment?
- ... In caption to (b), change "procedure described before" to "...procedure described in (a), ..."
- page 10, line 213. Refer to "nuclear thermal mixing" as "low-field thermal mixing" to immediately emphasize the importance of the 40 mT coil. It would also help to provide a field limit where mixing becomes a concern.
 - page 11, lines 226-8. The statement, "very little is known" is odd and open-ended. Change to something like, "No special measures were taken to optimize conditions for increased ¹³C lifetime. Higher field and lower temperature are known to increase the lifetime." And then continue with rest of the paragraph.
 - page 12, line 242. Add opening phrase to, "Following the 16-hour hold, the sample was then dissolved..."
 - page 12, line 248. Add opening phrase and change "time" to "times of" in, "In the dissolution product, we measured ¹³C spin-lattice relaxation times of T₁(¹³C) = ..."
 - page 12, line 250. Correct "fact the" to "fact that the"
 - page 13, Figure 4 caption. Better emphasize that this result is from a stored/transported sample by putting it alone and up front as in, "Hyperpolarized ¹³C NMR measured 16 hours after removal of the sample from the polarizer. Intervening storage was at 4.2 K and 1.0 T. Prior to storage, the mixture of [¹³C₃, ¹⁵N]Alanine and [¹³C₂, ¹⁵N]Glycine had been polarized at 6.7 T and 1.2 K. After storage, dissolution was achieved by standard means, but in the storage Dewar. Measurement was in an 11.7 T spectrometer (500 MHz for protons)."
 - page 14, line 272. The authors open discussion with, "This proof-of-concept experiment is far from optimal, and the levels of..." This comes across as overly humble, underselling the value of these results. I understand their meaning, that although polarization levels achieved after storage are good, not fantastic, there is much room to later improve towards 'fantastic' levels. Fine, but that point could be made a more elegantly without giving the reader the wrong impression about the quality of the present results.

Reviewer #2 (Remarks to the Author):

The paper describes a new DNP approach where the radical is spatially separated from the substance to be polarized in order to increase the relaxation times. This enables longer storage of polarized substances at liquid-Helium temperatures compared to samples where the radical and the polarization target are intimately mixed.

I think this is an interesting approach despite the fact that the achievable polarization levels are much lower than the ones obtained by direct polarization/dissolution. As the authors point out, there might be room for improvement but whether several orders of magnitude are possible is, at least in my opinion, doubtful. This is clearly a new approach but it is closely related (and inspired?) by the brute-force polarization approach of Kempf et al. at Bruker (ref 32/33). I think it is also related to the DNP of silicon nano particles where also the polarization generation on the surface is

spatially separated from the long-term storage of the polarization inside the silicon nano particles. Polarization transport is in this case also mediated by spin diffusion. I think this work by Cassidy should also be cited (M.C. Cassidy, H.R. Chan, B.D. Ross, P.K. Bhattacharya, C.M. Marcus, In vivo magnetic resonance imaging of hyperpolarized silicon particles, *Nature Nanotech.* 8 (2013) 363-368. doi:10.1038/nnano.2013.65., M.C. Cassidy, C. Ramanathan, D.G. Cory, J.W. Ager, C.M. Marcus, Radical-free dynamic nuclear polarization using electronic defects in silicon, *Phys. Rev. B.* 87 (2013). doi:10.1103/PhysRevB.87.161306. and maybe additional newer articles since I did not follow this topic in too much detail).

From the current draft it is difficult to judge how much polarization is obtained in the end since axis labels for the polarization are different. In Fig. 1 b and c it is absolute polarization which I think should be the axis for all the other figures. Fig. 1d suddenly has relative polarization and I think this needs to be converted into absolute polarization to make it comparable to b and c. Figure 2 should also have an absolute polarization given in the figure caption and also Figure 4b should have an absolute polarization axis and not epsilon where one is never sure how it is defined.

The polarization-loss plots of Fig. 2 show four samples. I would suggest that the authors add the two missing samples also to Fig. 4 to make this consistent.

The discussion section is too qualitative and the authors should discuss in more detail where the polarization losses occur and what possibilities there are to end up with polarization levels that are comparable to state-of-the-art dissolution DNP samples. For example, it looks like the polarization levels in the 1.2K solid are about a factor of 2 lower than what one can achieve in a glassy sample. But in the end one has about a factor of 100 (correct? difficult to estimate since no comparison and no absolute polarization levels are given) less than in a typical dissolution DNP experiment. Where does the main loss occur (transfer?, storage?). This is crucial to judge the potential of this method and to judge whether this is a nice experiment or an experiment that has high potential.

Reviewer #3 (Remarks to the Author):

The authors report on a novel methodological approach to obtain hyperpolarized ^{13}C with extremely long T_1 . The long T_1 characteristics allows to transport the hyperpolarized frozen substrate to a remote site where it can be exploited, after dissolution, for in-vivo MRI and other experiments.

The use of two phases, the micro-particulate substrate and the impregnating solution with the radical, is innovative and opens to new potential in research and applications. This is a very important step towards the routine use of hyperpolarized molecules in different areas and, in particular, at a clinical level.

The process occurs in 3 steps: first standard ^1H DNP occurs in a radical rich (CPA) region then, thanks to nuclear spin diffusion, the polarization is transferred to ^1H nuclei in a radical free (CCM) region and finally, after CP transferred to ^{13}C nuclei. Since ^{13}C nuclei are far away from the radicals their spin-lattice relaxation is particularly long, reaching up to 37 h, and driven by the lattice dynamics, which at low temperature are not much effective. The feasibility of this approach stems from the possibility to obtain micron size CCM regions surrounded by CPA ones, which can be removed after dissolution.

The manuscript is not only of interest for the NMR - DNP community but very relevant for the application of dissolution DNP in clinical diagnostics and in other areas and it deserves to be published.

Few comments:

-The samples were ground by hand, possibly with a quite irregular distribution of grain size, although from Figure 1 they appear all smaller than $1\mu\text{m}$. It would be important in the follow up of these experiments to control the grain size and investigate how the DNP efficiency changes with the grain size.

- While for the present proof-of-concept the authors use toluene as solvent of TEMPOL-benzoate, the challenge is to find suitable organic non-toxic solvents for all in-vivo medical application. A word of caution is probably necessary.

-Have the authors studied the temperature dependence of $1/T_1$ up to liquid nitrogen T? In some materials commonly used for DNP there is not a significant shortening of T_1 increasing the temperature up to 78 K, where less demanding storage conditions are required.

Reviewer #1 (Remarks to the Author):

This manuscript presents and demonstrates an innovative, potentially transformative method for the key emerging field of metabolic imaging of hyperpolarized nuclear spins. In 'conventional' applications in this field, hyperpolarization is typically generated by dynamic nuclear polarization (DNP) using a cryogenic, microwave polarizer that is situated very close to the imaging scanner. Dissolution of the sample from the cryostat, and then injection to a patient, allow immediate imaging. A desirable, but not previously possible variation, would be to instead remove several hyperpolarized samples from the polarizer in the frozen solid state, and then store them for later use and, especially, deliver them to far away imaging centers. That would avoid the complexity and cost of co-siting cryogenic polarizers and imaging scanners, thus bringing hyperpolarized metabolic MRI to a much broader base of researchers, practitioners and patients.

Before now, it has been thought that removing frozen solid samples from a DNP polarizer would not be possible, because DNP requires the presence of free-radical electrons. The radicals cause rapid loss of hyperpolarization in the otherwise slowly relaxing solid state. Here, the authors' innovated by preparing samples with microscale separation of two components: (a) the radical-embedded DNP medium, and (b) micro-crystallites of the metabolic compound to be hyperpolarized. Not only do the authors show that large hyperpolarization (~10%, or near 10,000x enhancement) can be achieved in several compounds of interest, they also demonstrate low-loss removal from the polarizing cryostat, and then more than a day or more of storage before eventual dissolution and magnetic resonance observation. This proof of concept is sure to inspire new approaches in this exciting field, and may also enable new applications (e.g., one possibility, multiple-injection hyperpolarized imaging from several hyperpolarized samples)

I am enthusiastic about publication of this work in Nature Communications. It meets the bar for high impact and broad interest. It will be valuable not only to the very large field of metabolic and cancer imaging, but also for applied spectroscopists, who might utilize the approach in studies of (e.g.) drug binding, protein-protein interactions, etc..., where one might simply order hyperpolarized substrates to enable a new kind of experiment on already widely used NMR equipment.

While I believe this paper should ultimately be accepted to Nature Communicaitons, I feel the authors must first address several concerns and clarifications. Significant issues follow here, after which I note minor issues.

Significant Questions / Changes:

(1) The authors should better comment on why the glycine/alanine mixture was chosen for the ultimate storage/transport experiment (Figure 4) as opposed to the other cases tested for removal from the polarizer (Figure 2). This is especially important since the authors had earlier (page 5, lines 17-121) emphasized the particular importance of pyruvate for imaging applications and gave it full focus in Fig 1 to explain and motivate the whole method. Also, they argued the CCM/CPA method should be general to any microcrystallite-forming molecule,

... based on Figure 2, it's clear that losses on removal were largest for pyruvate and smallest for Ala/Gly. So the motivation is pretty obvious. And yet explanation is still needed, including comments on why pyruvate suffered so much on removal from the cryostat.

In response to earlier work, we have been encouraged to broaden the scope of our research to encompass a variety of small molecules like amino-acids and sugars, rather than narrow the focus on Pyruvate. As the reviewer noted, the losses were found to be smaller for Ala and Gly than for Pyruvate. At this time, we do not have a satisfactory explanation for these differences in behavior. As expressed in the article page 11, we suspect that the methyl and possibly the quadrupolar sodium nuclear spin are involved in this relaxation loss at low magnetic fields and/or high temperature.

"For Pyruvate, the manual transfer strategy is not satisfactory, and a faster automated, cold, higher field transfer would help overcoming the fast relaxation arising from the presence of methyls and quadrupolar Sodium nuclear spins in the vicinity of the ^{13}C ."

(2) Starting on page 5, line 124, the authors refer to 'impregnation of the CCM by the CPA. This is misleading usage that should be dropped. The CCM is not impregnated by the CPA. Rather, isolated islands of CCM float in the CPA, and so it would be more accurate to say the CPA is impregnated by isolated CCM domains.

... the noted usage is consistent in the manuscript (e.g., page 6, lines 137-9, Figure 1 caption, and many others). So please search and correct all instances together.

The correct expression should be "wet impregnation", we have changed the text accordingly.

See for exemple: A Generic Wet Impregnation Method for Preparing Substrate-Supported Platinum Group Metal and Alloy Nanoparticles with Controlled Particle Morphology, Nano Lett., 2016, 16 (1), pp 164–169

(3) In Conclusions, the authors state "This approach enables for the first time the removal of the hyperpolarized solid from the polarizer while preserving polarization, and we have demonstrated for the

first time that storage or transport of hyperpolarized small molecules to remote locations using a simple cryogenic transport device is possible."

... This is not entirely true and the authors need to be more careful. Both noted firsts are only true if qualified for hyperpolarization "produced by DNP".

Elsewhere, the authors cited Hirsch, et al, JMR 2015, who hyperpolarized and transported pyruvic acid using the brute-force method. The authors must be equally clear in the Conclusions section. There is no reason to avoid the comparison there, i.e., it is clear that their enhancements are so far much bigger with DNP than using brute force. And so the actual 'firsts' of this paper are still high impact.

... This Conclusion must be rewritten more openly, and also with argument as to why this DNP approach may be best.

We have modified our Conclusions to read: "This approach enables the removal from the polarizer of a solid hyperpolarized by DNP while preserving its polarization. This is reminiscent of samples that have been hyperpolarized by so-called brute-force methods (i.e., at much lower temperatures without radicals^{32, 33 = Hirsch et al.}), although DNP can be much faster and can achieve much higher polarization levels."

(4) The title "Remote Hyperpolarization" lacks some meaning. I would prefer something more descriptive, such as "Transportable Hyperpolarization from Remote Dynamic Nuclear Polarization".

We changed it to "Transportable Hyperpolarized Metabolites"

(5) Comment: From Fig 1b, it appears that only about 15 min is needed for build-up to about 50% of the maximum achievable by traditional DNP of a homogeneous frozen solution. And then, another ~1 - 1.5 hrs are needed to get the final 50%. It would be interesting to know if the hyperpolarization obtained in the initial ~15 min has a similar or much shorter lifetime than the ultimate hyperpolarization obtained after 1-1.5 hrs.

At this time however, we have no experimental evidence that would allow us to correlate build-up times with lifetimes after the microwave irradiation has been switched off.

(6) In discussion the CP + spin diffusion method to buildup ¹³C hyperpolarization (page 6, lines 160-1), the authors wrote that two features of the method [(i) the longer time needed between CP contacts, and (ii) the (very) low loss during that time] imply that ¹³C polarization is slower to build-up, but can in principle achieve higher final polarization levels, theoretically up to $P(^{13}\text{C}) = P(^1\text{H})$.

That is indeed what we claimed.

... However, I don't see how the data supports that the final claim is distinct over prior methods. In Fig 1c, the traditional method (without CCM isolation) yielded a sawtooth buildup of $P(^{13}\text{C})$ that appears to asymptotically approach $P(^{13}\text{C}) \sim 50\text{-}60\%$. That is very similar to the ultimate 1H proton polarization that was shown in Fig. 1b. So it seems that the limit $P(^{13}\text{C}) \approx P(^1\text{H})$ can be obtained either with or without the new innovation.

In contrast to our earlier work, the polarization $P(^{13}\text{C})$ does not significantly drop in the intervals between CP contacts.

... The real key (properly noted by the authors and clear from their very good Fig 1c) is that the losses appear to be negligible in the CCM/CPA case, whereas they are rapid and large for DNP of a traditional formulation. It seems an overstatement to say that higher $P(^{13}\text{C})$ could be obtained using the CCM/CPA mixture. Better to say that the new method is "equally capable of ultimately achieving the $P(^{13}\text{C}) = P(^1\text{H})$ limit."

We have now reformulated our statement: "These new features imply that the ^{13}C polarization is slower to build-up, but can in principle achieve polarization levels as high as $P(^{13}\text{C}) = P(^1\text{H})$."

(7) A comment: when reading results, I immediately asked, "Can a faster buildup be obtained by using smaller CCM crystallites?" Much later (lines 276-280 in discussion), the authors take on this point and suggest several approaches to adjust particle size and possibly improve throughput. This is key discussion, and an earlier mention might be helpful. Please consider as part of your revisions.

We have inserted on page 6: "Smaller particle diameters reduce the distance over which hyperpolarized ^1H magnetization needs to diffuse and accelerate the build-up."

(8) The authors should also discuss another aspect of production throughput. That is, DNP is still a mostly one-sample-at-a-time method. Will that hold back development of a distributed system to deliver multiple 'doses' of hyperpolarization? What is the ultimate potential for multi-sample high throughput? ... Citations to Matthias Ernst and to GE's SpinLab as precedents for multisample production should be given. Elsewhere, they cite GE, but not Krajewski ... Ernst, Kozerke, "A multisample dissolution DNP system for serial injections in small animals.", *Mag.Res.Med.* (22 Feb 2016).

We have inserted a sentence in the conclusion, citing the following reference

Batel M, Krajewski M, Weiss K, With O, Dapp A, Hunkeler A, et al. A multi-sample 94 GHz dissolution dynamic-nuclear-polarization system. *J Magn Reson* 2012, **214**: 166-174

“With this new approach, the production of hyperpolarized molecules could in principle be scaled up, using dedicated high-throughput multiple-sample⁴¹ remote polarizers.”

Specific minor concerns / corrections:

- page 3, lines 64-65. Where authors write, "However, in all these cases paramagnetic relaxation in low fields precludes remote d-DNP as described below", which would be improved by two main changes:

(1) Please be a more explicit here, rather than waiting to the end of next paragraph for explanation. E.g., instead use, "However, in all these cases, intimate contact of the nuclear spins with the PA yields paramagnetic relaxation that... "

(2) This sentence should not end the paragraph, and would be better as the lead of the next paragraph. Along with that move, there should be some editing of the text for content and proper flow. Specifically, (a) dropping "precludes remote DNP" and instead writing that paramagnetic relaxation "requires dissolution of the sample directly in the cryostat."

'Remote DNP' is not really precluded, but rather is the authors' solution to the fact that the relaxation would normally preclude removal of the sample in the solid state.

(b) adding, "The resulting hyperpolarized solution has adequate lifetime ($T_1(^{13}\text{C}) \sim 30 - 60\text{s}$ in carboxyls) for immediate imaging or spectroscopy, but not for transport of the sample to a distant user site."

... and then there will be minor adjustments to remainder of the paragraph to avoid redundancies.

We have rewritten our text as follows: "However, intimate contact of the nuclear spins with the PA leads to paramagnetic relaxation that is exacerbated at low fields and thus requires dissolution of the sample directly in the cryostat. Hyperpolarized solutions have lifetimes $T_1(^{13}\text{C}) \sim 30 - 60\text{ s}$ in carboxyl groups that are sufficiently long for immediate imaging or spectroscopy, but not for transport of the sample to a distant user site"

page 4, line 77. Drop the word "would". The unnecessary subjunctive case weakens the authors' statement.

We have replaced "the PAs and the substrate would need" by "the PAs and the substrate need"

- page 4, lines 79-80. There is an awkward break from traditional methods, to the method of the authors. This is easily fixed by adding the word 'here' and using active voice, as in "However, here we achieve this by preparing DNP samples with a suitable multi-component..."

We have adopted this suggestion "However, here we achieve this by preparing DNP samples"

- page 4, lines 87-88. Change to "Here, we show that particles with diameters on the order of a few μm typically result in a timescale of 30 minutes for proton-proton spin diffusion."

... The original combined "in a timescale with "on the order of", which is an unnecessary double qualification.

... Also, "timescale of 30 min" seems fair (e.g., time constant of 1900 s noted on line 141) and better quantifies for the reader.

We have adopted: "in a time-scale of 30 min for proton-proton spin diffusion"

... Finally, the authors might comment if they expect any field or temperature dependence for this timescale given that polarizer conditions may vary.

We hesitate to speculate on any field or temperature dependence, by lack of empirical observations

page 4, line 89. Change to "...from 1H spins in the CCM to 13C spins of the metabolites."

We have adopted this suggestion

page 4, line 90. The authors wrote, "At this point, DNP is switched off, and...", It is not clear if they mean that DNP are switched off before or just after cross polarization. This should be clarified.

We have clarified this as follows:

"After the ^{13}C polarization has achieved a satisfactory level, the microwave irradiation is switched off and the sample temperature is increased from 1.2 K to 4.2 K (ambient pressure), and the protons in both radical-rich phase (RRP) and radical-free phase (RFP) relax towards thermal equilibrium within minutes"

Also, in Supporting Information, experiment timelines should be included, e.g., one pulse sequence (rf + microwaves + diffusion delays, etc...) should be given corresponding to Figures 1b and 1c.,

We have added this information in a supplement.

page 4, line 92. Typical should not be used without defining the corresponding conditions. Please change "...can remain polarized, typically for days." to, "..., typically for days when at polarizer conditions of 6.7 T, and 1.2 K."

We have specified:" typically for several days at 4.2 K in a field of 6.7 T"

page 4, line 96. Change "...13C-13C spin diffusion is ineffective in the CPA." to "...13C-13C spin diffusion is ineffective in the natural-isotopic-abundance CPA." This is not essential, but it does make it easier for the reader to immediately grasp the reason.

We have specified: " ^{13}C - ^{13}C spin diffusion is intrinsically two orders of magnitude slower than ^1H - ^1H spin diffusion in the RFP, and is ineffective in the RRP since it is not isotopically ^{13}C enriched"

- page 4, line 97. Consider using "spin-diffusion assisted, relayed DNP"

We prefer the formulation "the concept of DNP relayed by spin diffusion"

- page 5, line 103. "magic angle spinning DNP (MAS)" to "magic angle spinning (MAS) DNP"

We have adopted this suggestion

page 5, line 121-22. Add reference to SEM and/or other method used, e.g., "Here, powdered samples were ground by hand to an average particle diameter of $1 < d < 10 \mu\text{m}$, as determined by SEM. (See Figure 1a.)"

We have inserted "as determined by SEM"

On page 5, line 123, the CPA description should start by just giving the composition of the CPA. As written, It was strange to start a section on CPA with "The CCM made of micro-particles is impregnated by a CPA consisting of...". Also, see earlier note on 'impregnation'.

We have now reformulated:

"The radical rich phase (RRP) consists of a glass-forming solution of polarizing agent (PA) chosen such that the RFP is not soluble in the RRP, such as 80 mM TEMPOL-benzoate in Toluene/THF (v:v = 8:2)."

On page 6, line 126, The authors should expand "(see methods)" to "(See methods for details on combining CCM and CPA, including procedures to homogenize the distribution, and to fix the sample by freezing.)". The later bit on fixing is not currently present, but would be a simple, useful addition.

We have adopted this suggestion.

- page 6, lines 148-9. Change to "was performed here" instead of "can be performed" in "..., cross-polarization from ^1H to ^{13}C by CP-DNP was performed in essentially the same manner."

We have adopted this suggestion.

- page 6, lines 137-9. Where it states, "The red curve in Figure 1b shows the ^1H DNP build-up obtained under the same conditions by impregnating 20 mg of the microcrystalline sodium $[1-^{13}\text{C}]$ pyruvate CCM sample with the CPA."

... The authors should specify the fractional volume contributions of CPA and CPM, as well as the proton densities in each.

We have given the requested information

This can later be pulled out of Methods, but would be better also given here.

... and again, fix 'impregnation'.

We have specified "wet impregnation"

- page 8, Figure 1.

... The labeling text in 1b should be updated to match acronym definitions, i.e., change to "component containing polarizing agents (CPA) and "component containing metabolites (CCM)"

We have adopted "radical rich phase" (RRP) (instead of our clumsy CPA) and "radical free phase" (RFP) (to replace CCM). See reviewer 3's first comment.

... In caption for (b), state "Curves are single and bi-exponential fits to the data, as described in the text."

We have adopted this suggestion.

... In caption for (c), add reference at "several CP pulse sequences are applied[REF] to transfer..." Also, please refer to sequences newly provided in Supp. Info, as requested above.

We have inserted a reference to the revised Supplement.

page 9, line 184. Add "In contrast," to "In contrast, the ^{13}C spins of metabolites dissolved in glassy..."

We have inserted:

"in contrast to ^{13}C spins of metabolites dissolved in conventional glassy DNP samples"

- page 8, Figure 2, and corresponding discussion.

... On page 10, line 211-14, the authors wrote that "Special care needs to be taken with regard to the magnetic field trajectory for sample shuttling out of the polarizer... Figure 2 shows how a manual sample lift above the polarizer..." The whole intent of Fig 2 results would be clearer if the second sentence started with "In order to test the feasibility of this first step towards transportable hyperpolarization, Figure 2 shows..."

We have inserted "In order to test the feasibility of our strategy towards transporting hyperpolarization,"

...The 2nd sentence of the caption to part (a) is confusing. It should be broken into at least two sentences, with some clarifying changes. E.g., change to "In practice, over a period of about 15s, the sample is lifted up (____) 40 mT supplementary field coil (____) 90 cm above the center of the main B_0 field. The supplementary field is in the same direction as B_0 field and limits sample exposure to low field."

The first and second blanks (____) should be filled with either "while within a" and "to a distance of", OR "and into" and "located at". That depends on whether the 40 mT coil moved with the sample OR if the sample moved into it. I assume the former, but if not, what was the minimum exposure while moving?

It seems the 40 mT coil here was probably the same one noted in Fig 3(c), but that is not clear with Fig. 2. If the same, then the authors should modify Fig 2 to incorporate Fig. 3(c) as an inset and also describe it in the caption. (They should also still keep this picture in Fig.3(c).)

... The authors also wrote "..., where the sample is in a room temperature environment, and is put back in the polarizer in approximately the same amount of time."

- please add "for about 30 s" to the first phrase and change the last phrase to "in another ~15 s."

- Is there a special reason to come to room T, rather than colder, polarization-preserving environment?

... In caption to (b), change "procedure described before" to "...procedure described in (a), ..."

We have rewritten the caption to Fig 2 as follows: "**Figure 2.** (a) Procedure for lifting the sample above the polarizer and putting it back. In practice, the sample is lifted manually by 2 cm into a coil that provides a supplementary field of 40 mT, parallel to the B_0 field of the polarizer. In 5 s, the sample and the supplementary coil are then lifted together 90 cm above the center of the main B_0 field. The sample is kept at room temperature for about 30 s and is put back into the polarizer in approximately 5 s. (b) ^1H decoupled ^{13}C NMR spectra of [$^{13}\text{C}_2$, ^{15}N]Glycine, [$^{13}\text{C}_3$, ^{15}N]Alanine, [$1\text{-}^{13}\text{C}$]Glucose, and [$1\text{-}^{13}\text{C}$]Pyruvate measured with 0.5° nutation angle pulses after 10 min ^1H DNP followed by a single $^1\text{H}\text{-}^{13}\text{C}$ cross polarization contact, before (red line) and after (blue line) lifting the sample according to the procedure described in (a), leading to polarization losses of ca. -8%, -13%, -15%, and -70% respectively."

- page 10, line 213. Refer to "nuclear thermal mixing" as "low-field thermal mixing" to immediately emphasize the importance of the 40 mT coil. It would also help to provide a field limit where mixing becomes a concern.

We have replaced "nuclear thermal mixing" by "low-field nuclear thermal mixing". We do not wish to speculate on the field below which mixing becomes effective

- page 11, lines 226-8. The statement, "very little is known" is odd and open-ended. Change to something like,

"No special measures were taken to optimize conditions for increased ^{13}C lifetime. Higher field and lower temperature are known to increase the lifetime." And then continue with rest of the paragraph.

We have reformulated the text as follows: "No attempts were made so far to optimize the ^{13}C lifetimes during the waiting time of 16 hours. Higher fields, lower temperatures, sample purification, oxygen removal, and annealing of the RFP are expected to improve the lifetimes, as shown by Kempf *et al.*..."

page 12, line 242. Add opening phrase to, "Following the 16-hour hold, the sample was then dissolved..."

We have adopted "Following a 16-hour hold, the ..."

page 12, line 248. Add opening phrase and change "time" to "times of" in, "In the dissolution product, we measured ^{13}C spin-lattice relaxation times of $T_1(^{13}\text{C}) = \dots$ "

We have adopted this suggestion.

- page 12, line 250. Correct "fact the" to "fact that the"

This correction has been made.

page 13, Figure 4 caption. Better emphasize that this result is from a stored/transported sample by putting it alone and up front as in, "Hyperpolarized ^{13}C NMR measured 16 hours after removal of the sample from the polarizer. Intervening storage was at 4.2 K and 1.0 T. Prior to storage, the mixture of [$^{13}\text{C}_3$, ^{15}N]Alanine and [$^{13}\text{C}_2$, ^{15}N]Glycine had been polarized at 6.7 T and 1.2 K. After storage, dissolution was achieved by standard means, but in the storage Dewar. Measurement was in an 11.7 T spectrometer (500 MHz for protons)."

We have rewritten the caption to Fig. 4: "(a) Hyperpolarized ^{13}C NMR signals measured 16 hours after removal of the sample from the polarizer. The sample was stored at 4.2 K and 1.0 T. Prior to storage, 5.5 mg [$^{13}\text{C}_3$, ^{15}N]Alanine and 9 mg [$^{13}\text{C}_2$, ^{15}N]Glycine were mixed with 45 μL of Toluene:THF (8:2) doped with 80 mM TEMPOL-benzoate (RRP) and polarized at 6.7 T and 1.2 K. Following a 16 hour hold, the sample was dissolved..."

page 14, line 272. The authors open discussion with, "This proof-of-concept experiment is far from optimal, and the levels of..." This comes across as overly humble, underselling the value of these results. I understand their meaning, that although polarization levels achieved after storage are good, not fantastic, there is much room to later improve towards 'fantastic' levels. Fine, but that point could be made a more elegantly without giving the reader the wrong impression about the quality of the present results.

We have removed some excessively humble statements

Reviewer #2 (Remarks to the Author):

The paper describes a new DNP approach where the radical is spatially separated from the substance to be polarized in order to increase the relaxation times. This enables longer storage of polarized substances at liquid-Helium temperatures compared to samples where the radical and the polarization target are intimately mixed.

I think this is an interesting approach despite the fact that the achievable polarization levels are much lower than the ones obtained by direct polarization/dissolution. As the authors point out, there might be room for improvement but whether several orders of magnitude are possible is, at least in my opinion, doubtful.

Indeed, it appears doubtful that we can gain more than one order of magnitude in polarization

This is clearly a new approach but it is closely related (and inspired?) by the brute-force polarization approach of Kempf et al. ab Bruker (ref 32/33).

We have given more credit to Kempf et al., whose work has indeed been a source of inspiration for our strategy.

I think it is also related to the DNP of silicon nano particles where also the polarization generation on the surface is spatially separated from the long-term storage of the polarization inside the silicon nano particles. Polarization transport is in this case also mediated by spin diffusion. I think this work by Cassidy should also be cited (M.C. Cassidy, H.R. Chan, B.D. Ross, P.K. Bhattacharya, C.M. Marcus, In vivo magnetic resonance imaging of hyperpolarized silicon particles, Nature Nanotech. 8 (2013) 363-368. doi:10.1038/nnano.2013.65.,M.C. Cassidy, C. Ramanathan, D.G. Cory, J.W. Ager, C.M. Marcus, Radical-free dynamic nuclear polarization using electronic defects in silicon, Phys. Rev. B. 87 (2013).

doi:10.1103/PhysRevB.87.161306. and maybe additional newer articles since I did not follow this topic in too much detail).

We have inserted references to Cassidy et al. and modified the text: “in contrast to ^{13}C spins of metabolites dissolved in conventional glassy DNP samples and similar to silicon nanoparticles”

From the current draft it is difficult to judge how much polarization is obtained in the end since axis labels for the polarization are different. In Fig. 1 b and c it is absolute polarization which I think should be the axis for all the other figures. Fig. 1d suddenly has relative polarization and I think this needs to be converted into absolute polarization to make it comparable to b and c. Figure 2 should also have an absolute polarization given in the figure caption and also Figure 4b should have an absolute polarization axis and not epsilon where one is never sure how it is defined.

For figure 1d we did not measure the absolute ^{13}C polarization in the polarizer at low temperature because T_1 is extremely long so that we would have had to wait several days to measure a thermal equilibrium signal. The same applies to figure 2. In both cases it is the relative loss of polarization that matters, rather than the absolute polarization.

What matters in the end is the final polarization in the liquid after dissolution. We have therefore modified figure 4 as suggested by the reviewer.

The polarization-loss plots of Fig. 2 show four samples. I would suggest that the authors add the two missing samples also to Fig. 4 to make this consistent.

We have not performed any remote dissolution experiments on the two other samples.

The discussion section is too qualitative and the authors should discuss in more detail where the polarization losses occur and what possibilities there are to end up with polarization levels that are comparable to state-of-the-art dissolution DNP samples. For example, it looks like the polarization levels in the 1.2K solid are about a factor of 2 lower than what one can achieve in a glassy sample. But in the end one has about a factor of 100 (correct? difficult to estimate since no comparison and no absolute polarization levels are given) less than in a typical dissolution DNP experiment. Where does the main loss occur (transfer?, storage?). This is crucial to judge the potential of this method and to judge whether this is a nice experiment or an experiment that has high potential.

We have attempted to provide answers to this legitimate concern by inserting estimates of the expected losses in the discussion. We do not currently have the tools in hand to foresee the maximum polarization

that we will be able to achieve in the future. It seems impressive that we are only one order of magnitude lower than the levels of polarization typically used in imaging.

Reviewer #3 (Remarks to the Author):

The authors report on a novel methodological approach to obtain hyperpolarized ^{13}C with extremely long T_1 . The long T_1 characteristics allows to transport the hyperpolarized frozen substrate to a remote site where it can be exploited, after dissolution, for in-vivo MRI and other experiments.

The use of two phases, the micro-particulate substrate and the impregnating solution with the radical, is innovative and opens to new potential in research and applications. This is a very important step towards the routine use of hyperpolarized molecules in different areas and, in particular, at a clinical level.

The process occurs in 3 steps: first standard ^1H DNP occurs in a radical rich (CPA) region then, thanks to nuclear spin diffusion, the polarization is transferred to ^1H nuclei in a radical free (CCM) region and finally, after CP transferred to ^{13}C nuclei. Since ^{13}C nuclei are far away from the radicals their spin-lattice relaxation is particularly long, reaching up to 37 h, and driven by the lattice dynamics, which at low temperature are not much effective. The feasibility of this approach stems from the possibility to obtain micron size CCM regions surrounded by CPA ones, which can be removed after dissolution.

The manuscript is not only of interest for the NMR - DNP community but very relevant for the application of dissolution DNP in clinical diagnostics and in other areas and it deserves to be published.

We have adopted “radical rich phase” (RRP) (instead of our clumsy CPA) and “radical free phase” (RFP) (to replace CCM)

Few comments:

-The samples were ground by hand, possibly with a quite irregular distribution of grain size, although from Figure 1 they appear all smaller than $1\ \mu\text{m}$. It would be important in the follow up of these experiments to control the grain size and investigate how the DNP efficiency changes with the grain size.

Indeed. We are currently working on methods to obtain mono-dispersed powders.

While for the present proof-of-concept the authors use toluene as solvent of TEMPOL-benzoate, the challenge is to find suitable organic non-toxic solvents for all in-vivo medical application. A word of caution is probably necessary.

We have inserted a word of caution about toluene: “We have not yet investigated bio-compatible solvents that could replace Toluene.”

-Have the authors studied the temperature dependence of $1/T_1$ up to liquid nitrogen T ? In some materials commonly used for DNP there is not a significant shortening of T_1 increasing the temperature up to 78 K, where less demanding storage conditions are required.

We hesitate to speculate about parameters that we cannot determine with our current instrumentation. If $T_1(^{13}\text{C})$ were the same at 70 and 4 K, this would greatly simplify storage and transport.

REVIEWERS' COMMENTS:

Reviewer #1 (Remarks to the Author):

The authors have adequately addressed all issues raised by me and by other reviewers. (I especially like the changes suggested by another reviewer to use "RRP" and "RFP".)

I recommend immediate publication.

First, one simple remaining request. In first review the authors responded to one of my comments saying, "We do not wish to speculate on the field below which mixing becomes effective."

This is *mostly* acceptable, however recent literature (Peat, et al) shows that low-field thermal mixing (for at least some samples) is inactive above 20 mT. The sample there was neat pyruvic acid, similar to pyruvate crystallites, one of the four samples studied here.

Granted that offers a guideline/ballpark figure, not exact comparison, so the authors' hesitation to speculate is understandable. Nonetheless, citation and comment are warranted on page 11, lines 230-232, as in

"Special care needs to be taken with regard to the magnetic field trajectory for sample shuttling out of the polarizer, so as to avoid excessive relaxation losses and low-field nuclear thermal mixing (36, 37), as Peat, et al recently characterized in a similar sample type, neat [1-13C] pyruvic acid [REF]."

[REF] = Peat, et al PCCP, 2016,18, 19173-19182

Reviewer #2 (Remarks to the Author):

I think the authors have responded well to the comments of the reviewers and have included most of the suggestions. As far as I am concerned the article can be published in its current form.

Reviewer #3 (Remarks to the Author):

The revised version of the manuscript incorporates all specific and general comments made in the previous Comments for Authors. Furthermore, I have been delighted to go through the Remarks provided by the other two Reviewers and the Authors' reply. I believe the manuscript fits all the requirements for publication.

Please find enclosed our answer to the reviewer's comment

"Nonetheless, citation and comment are warranted on page 11, lines 230-232, as in

"Special care needs to be taken with regard to the magnetic field trajectory for sample shuttling out of the polarizer, so as to avoid excessive relaxation losses and low-field nuclear thermal mixing (36, 37), as Peat, et al recently characterized in a similar sample type, neat [1-13C] pyruvic acid [REF]."

[REF] = Peat, et al PCCP, 2016,18, 19173-19182"

we have added the reference and modified the text as follows:

"Special care needs to be taken with regard to the magnetic field trajectory for sample shuttling out of the polarizer, so as to avoid excessive relaxation losses and low-field nuclear thermal mixing ^{36,37} as recently illustrated by Peat et al.²⁶"